# SinGLU: Sinusoidal Gated Linear Units Improve Classification Accuracy of Small Vision Transformers

**Luke Byrne**                                              *luke.byrne@strath.ac.uk*
*Department of Electronic & Electrical Engineering*
*University of Strathclyde*
*Glasgow G1 1XW, United Kingdom*

**Paul Murray**                                            *paul.murray@strath.ac.uk*
*Department of Electronic & Electrical Engineering*
*University of Strathclyde*
*Glasgow G1 1XW, United Kingdom*

**Reviewed on OpenReview:** *https://openreview.net/forum?id=qq4yipldw2*

## Abstract

Gated Linear Unit (GLU) variants such as SwiGLU are now widely used in modern Transformers. However, the GLU functions explored in the recent literature represent only a small fraction of the possible GLU design space. Starting from a systematic enumeration of a restricted family of zeroth-, first-, and second-order GLU-type formulas, we conduct a controlled study on ViT-Tiny across CIFAR-10, CIFAR-100, SVHN and ImageNet64, instantiating each GLU formula with Sigmoid, Tanh and Sin activations. Under identical training recipes and matched parameter counts, our proposed first-order variant **SinGLU** achieves higher mean accuracy than SwiGLU across the datasets tested in this ViT-Tiny setting. Inference latency differs by <0.1% on an NVIDIA A100 GPU, confirming cost parity.

## 1 Introduction

Vision Transformers (Dosovitskiy et al., 2020) have seen widespread and rapid adoption across the field of computer vision. This is owed largely to their improved performance over previous machine learning architectures like Convolutional Neural Networks (CNNs) when large volumes of data are available. However, the Transformer architecture is surprisingly simple. A Vision Transformer first converts an image into a sequence of patch embeddings, adds positional embeddings, and processes the resulting token sequence using repeated Transformer encoder blocks. Each encoder block typically contains a Multi-Head Self-Attention (MHSA) module and a channel-wise MLP module, together with residual connections and Layer Normalization (Ba et al., 2016). Important nonlinear operations in a standard Transformer include the softmax normalization in attention, Layer Normalization, and the activation or gating function in the MLP block. This paper focuses on the last of these: the nonlinear layer function used inside the Transformer MLP.

The original Vision Transformer (Dosovitskiy et al., 2020) was proposed with a GELU (Hendrycks & Gimpel, 2016) activation as the nonlinearity in the first layer of the MLP. Recently Shazeer (2020) introduced a number of Gated Linear Unit (GLU) (Dauphin et al., 2017) type functions which outperformed GELU when used in the MLP portions of a language Transformer. Shazeer's SwiGLU function has since been used in several state-of-the-art large language and Vision Transformer models such as PaLM (Chowdhery et al., 2023), LLaMA (Touvron et al., 2023), and DINOv2 (Oquab et al., 2023). In Shazeer's paper SwiGLU was presented as a GLU-type layer with a Swish (Ramachandran et al., 2017) activation function.

---

Code: `https://github.com/Luke-Byrne-Eng/SinGLU-Experimental-Code`

The characteristics of a 'GLU-type' function are not strictly defined. The original GLU uses two affine projections, producing two groups of hidden neurons. One group of hidden neurons has their outputs passed through a nonlinear activation function while the other does not. The outputs of the two groups are then multiplied together element-wise, giving the layer's final output. In this way the activated nonlinear neurons can be seen as 'gating' the linear neurons, hence the name 'Gated Linear Unit'. Shazeer's variants similarly use separate projection matrices for the gated and value pathways, but omit bias terms in the affine projections.

In the original GLU, the activated and non-activated pathways use separate affine projections. Shazeer's GLU variants similarly use separate projection matrices for the gated and value pathways, but their implemented T5-style variants omit bias terms. In this work, we use a generalized affine notation with optional biases.

If we allow the affine projections to either be separate or identical then we can actually view the Swish 'activation' (Ramachandran et al., 2017) inside SwiGLU as itself being a GLU-type function with a Sigmoid activation. In this way SwiGLU can be seen as a *second-order* GLU-type function with a Sigmoid activation. It also becomes clear that other second-order variants are possible by changing which weight matrices are identical and which are unique. Visual comparisons between Transformer MLP blocks utilizing zeroth-, first-, and second-order GLU-type functions may be seen in Figure 1.

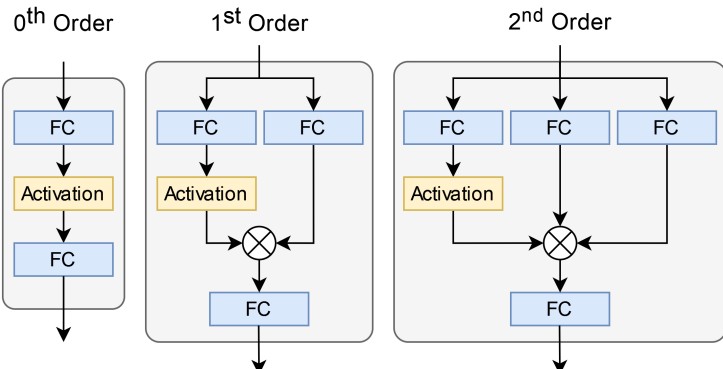

Figure 1: Block diagrams of Transformer MLP blocks with $0^{th}$, $1^{st}$, and $2^{nd}$ order GLU-type functions. Here 'FC' refers to a fully connected layer of neurons, 'Activation' represents an element-wise activation function, and $\otimes$ is an element-wise multiplication.

We therefore perform a controlled evaluation of a restricted family of GLU-type layer functions, varying both the order of the multiplicative interaction and the activation function used in the gated pathway. We restrict the search space to only functions with 1 to 3 affine projections. This reduces the number of possible $0^{th}$ to $2^{nd}$ order functions for a given activation to 7.

We choose Sigmoid, Tanh, and Sin as the activation functions investigated in this study because they span several relevant activation properties while remaining simple and interpretable. Sigmoid is monotonic and bounded to $[0, 1]$, and is the canonical gating activation in the original GLU. Tanh is monotonic, odd, zero-centred, and bounded to $[-1, 1]$. Sin is also odd and bounded to $[-1, 1]$, and satisfies $\sin(x) \approx \tanh(x)$ near zero, but differs from Tanh by becoming non-monotonic outside this region. Including Sin therefore allows us to test whether the normalization and initialization regimes used in modern Transformers make sinusoidal gating viable, despite the historical difficulty of training networks with periodic activations. Thus, the motivation for SinGLU is not merely to replace SwiGLU's activation with Sin, but to test whether a smooth, bounded, non-monotonic gate provides useful feature modulation when multiplied by a linear value pathway.

With 7 GLU-type layer functions and 3 activations, we create 21 ViT-Tiny (Wu et al., 2022) models, and assess their performance on a variety of image classification datasets.

Our experiments suggest that a first-order GLU-type layer with a Sin activation is a strong alternative to SwiGLU in the ViT-Tiny setting, particularly on the more densely sampled datasets considered here. We choose to call this function "SinGLU" in line with the naming conventions present in the literature.

The main contributions of this work are:

- A systematic enumeration of affine-projection sharing patterns within a restricted family of $0^{th}$ to $2^{nd}$ order GLU-type layer functions.

- Identification of several GLU-type layer functions that achieve higher mean accuracy than SwiGLU under matched ViT-Tiny training conditions.

- Evidence that sinusoidal activations can be used inside GLU-type Transformer MLP layers without obvious optimization failure under the training recipes considered.

## 2 Background and Related Work

### 2.1 Vision Transformers

The Vision Transformer (ViT) was introduced by Dosovitskiy et al. (2020). It built on the Transformer architecture proposed by Vaswani et al. (2017) for natural language processing. The Transformer encoder used in ViT processes a sequence of patch embeddings with positional embeddings through repeated encoder layers. Each encoder layer contains a Multi-Head Self-Attention module and a channel-wise MLP module, together with residual connections and Layer Normalization.

Analysis by Park & Kim (2022) has shown that the MLP and MHSA blocks perform opposite, complementary roles. The MLP acts as a high-pass filter and amplifier, increasing the variance of feature maps. The MHSA block acts as a low-pass filter, aggregating feature maps and reducing variance. This causes the MHSA to have a smoothing effect on the loss landscapes of ViTs, reducing the Hessian maximum eigenvalues, on sufficiently large datasets. The same analysis also showed that the MHSA block makes ViTs far less likely to overfit on small datasets. However, it is important to note that the lack of strong inductive biases in ViTs can result in less convex loss landscapes for small datasets. Tuli et al. (2021) have shown that the aggregating property of the MHSA substantially increases the network's bias towards shape rather than texture, when compared to ResNets (He et al., 2016). This bias aligns more closely with human perception and improves error consistency and robustness to domain shift, as measured by accuracy on Stylized-ImageNet (SIN) (Geirhos et al., 2018).

### 2.2 Gated Linear Units

The original gated linear unit (GLU) was introduced by Dauphin et al. (2017) for use in convolutional language modelling. The output of a GLU layer is the Hadamard, or element-wise product of the outputs of two fully connected groups of neurons, one of which has a Sigmoid activation function. Given an input feature map $x$, Sigmoid function $\sigma$, learnable weight matrices $W_1$, $W_2$, and learnable bias vectors $b_1$, $b_2$ the output of a GLU layer may be defined as:

$$GLU(x, W_1, W_2, b_1, b_2) = \sigma(xW_1 + b_1) \otimes (xW_2 + b_2) \tag{1}$$

where $\sigma$ is the Sigmoid function. Shazeer (2020) later introduced GLU variants of several popular activation functions, by replacing the Sigmoid with other activation functions such as Swish (Ramachandran et al., 2017) and RELU (Fukushima, 1969).

### 2.3 Online Normalization

We suspect that online normalization techniques may be beneficial when training neurons with periodic activation functions. Online normalization standardizes the data distribution before applying a learned

affine transformation. Several online normalization techniques exist, for example: Batch Normalization (Ioffe & Szegedy, 2015), Instance Normalization (Ulyanov et al., 2016), Group Normalization (Wu & He, 2018), etc.

Layer Normalization (Ba et al., 2016) is the technique most commonly used in Transformers. It normalizes each channel separately before applying a shared linear affine. It is especially suited for Transformer architectures, where fully connected layers in MHSA and MLP blocks act along the channel dimension only. A key design choice in Transformers is whether to apply Layer Normalization before or after these blocks: Pre-Layer-Normalization applies it at the block input, while Post-Layer-Normalization applies it at the output (Xiong et al., 2020). This paper focuses on models using Pre-Layer-Normalization. Recent research (Zhu et al., 2025) shows that large activation outliers can cause Layer Normalization to behave like a saturating Tanh function, especially in deeper layers and later training stages, where such outliers are more common.

### 2.4 Sinusoidal Activations in Pre-Norm Transformers

Periodic activation functions have seen limited adoption in mainstream neural network architectures, although they have been used successfully in more specialized settings such as Fourier Neural Networks (Ngom & Marin, 2021) and Implicit Neural Representations (Sitzmann et al., 2020). One reason for their limited use is that periodic activations are non-monotonic, which can introduce repeated local structure into the optimization landscape when pre-activations span many periods of the function (Parascandolo et al., 2016).

Transformers provide a more favourable setting for sinusoidal activations than the architectures considered by Parascandolo et al. (2016). Layer Normalization bounds the inputs to the MLP, and standard initialization schemes place most pre-activations near zero early in training. In this near-zero regime $\sin(x) \approx \tanh(x)$, so the sinusoidal gate is effectively monotonic at initialization and the optimization pathologies associated with periodic activations on unbounded inputs are largely avoided. As training progresses and weights grow, the gate can begin to exploit the non-monotonic structure of sin, modulating the value pathway in ways that monotonic gates such as Sigmoid or Tanh cannot.

This motivates evaluating Sin within GLU-type Transformer MLP layers. We do not assume that sinusoidal activations are universally easy to optimize, nor that SinGLU itself is periodic as a function of its input. Rather, we test whether the normalization and initialization regimes used in modern Vision Transformers make sinusoidal gating viable under standard training recipes.

## 3 Methods

### 3.1 GLU-type Layer Functions

Shazeer (2020) defined SwiGLU as:

$$SwiGLU(x, W, V, \beta) = Swish_{\beta}(xW) \otimes (xV) \tag{2}$$

Where $x$ is the input feature map, $W$ and $V$ are weight matrices, $\otimes$ is a Hadamard or element-wise product, and $\beta$ is a scalar parameter which is set to 1 in all of Shazeer's experiments. It should be noted that in Shazeer's formulation of GLU functions they removed the bias terms following the implementation of the T5 transformer architecture.

We include bias terms in our generalized notation because the original GLU formulation contains biases and because biases are relevant when analysing the phase and amplitude effects of sinusoidally gated neurons. In our notation, Shazeer's bias-free variants are recovered by setting the corresponding bias vectors to zero or omitting them.

When $\beta = 1$ Swish is equivalent to the SiLU activation of Hendrycks & Gimpel (2016), which they defined as:

$$SiLU(x) = \sigma(x) \otimes x \tag{3}$$

Where $\sigma$ is a Sigmoid function. Therefore, a generalized biased form of SwiGLU can be written as:

$$\text{SwiGLU}(x, W, V, b, c) = \sigma(xW + b) \otimes (xW + b) \otimes (xV + c). \tag{4}$$

We simplify this notation by defining each affine projection as:

$$x_n = xW_n + b_n \tag{5}$$

The output $Z$ of a biased SwiGLU layer can then be written as:

$$Z = \sigma(x_1) \otimes x_1 \otimes x_2 \tag{6}$$

Under our ordering convention, the order of a GLU-type layer is determined by the number of affine projection factors participating in the multiplicative interaction. SwiGLU is therefore a second-order GLU-type layer with a Sigmoid gate, since it contains the two affine factors $x_1$ and $x_2$ in addition to the activation $\sigma(x_1)$. This also makes clear that SwiGLU is one member of a broader family of second-order GLU-type layer functions.

### 3.2 Restricted Enumeration of GLU-type Layers

We seek to evaluate a broad but restricted family of GLU-type layer functions in order to compare how activation choice and multiplicative order affect performance under matched training conditions. As the space of all possible GLU-type layer functions is infinite, we restrict our search to an extensive but tractable range. We explore one to three $x_n$ terms, allow the activation $\phi$ to act on only one $x_n$ term, and allow only Hadamard product operations between $x_n$ terms. This range encompasses regular non-GLU layer functions, the original GLU, and the second-order function SwiGLU. However, this still leaves a wide range of functions unexplored, which should be investigated in future work.

Under these constraints the list of functionally unique $0^{th}$ to $2^{nd}$ order GLU-type layers with activation function $\phi$ is:

$$
\begin{array}{ll}
\underline{Order} & \underline{Layer\,Function} \\
0^{th} & Z_1 = \phi(x_1) \\
1^{st} & Z_2 = \phi(x_1) \otimes x_1 \\
1^{st} & Z_3 = \phi(x_1) \otimes x_2 \\
2^{nd} & Z_4 = \phi(x_1) \otimes x_1 \otimes x_1 \\
2^{nd} & Z_5 = \phi(x_1) \otimes x_2 \otimes x_2 \\
2^{nd} & Z_6 = \phi(x_1) \otimes x_1 \otimes x_2 \\
2^{nd} & Z_7 = \phi(x_1) \otimes x_2 \otimes x_3
\end{array} \tag{7}
$$

It should be noted that the zeroth-order case $Z_1 = \phi(x_1)$ is not a gating mechanism because it contains no multiplicative value pathway. We include it as a standard activation-function baseline, allowing us to separate the effect of the activation itself from the effect of multiplicative gating.

Although some higher-order variants may appear superficially related to their lower-order counterparts, they are not redundant in practice. For instance, $Z_2 = \phi(x_1) \otimes x_1$ and $Z_4 = \phi(x_1) \otimes x_1 \otimes x_1$ share the same affine projection inside and outside the gate, but the hidden dimension equalization (Eq. 9) assigns them different effective widths to match parameter count, and the additional Hadamard factor in $Z_4$ introduces a quadratic term in $x_1$ that alters the gradient flow through the gate. Similar reasoning applies to $Z_3 = \phi(x_1) \otimes x_2$ versus $Z_5 = \phi(x_1) \otimes x_2 \otimes x_2$. Variants with more independent projections introduce further bilinear interactions: expanding the non-activated component of $Z_7 = \phi(x_1) \otimes x_2 \otimes x_3$ gives

$$(xW_2 + b_2) \otimes (xW_3 + b_3),$$

which contains pairwise multiplicative interactions between independent input-dependent terms. This increased expressivity may benefit densely sampled datasets, but may also increase sensitivity to optimization and overfitting.

When $\phi = $ Sigmoid, $Z_1 = $ Sigmoid, $Z_2 = $ Swish$_{\beta=1}$, $Z_3 = $ GLU, and $Z_6 = $ SwiGLU. However, as far as we are aware, $Z_4$, $Z_5$, and $Z_7$ have not yet been explored. To the best of our knowledge $Z_4$ through $Z_7$ have never been explored for Tanh or Sin activation functions either.

This notation also recovers GELU and GEGLU if $\phi$ is chosen to be the standard Gaussian cumulative distribution function $\Phi$. Since

$$\mathrm{GELU}(x) = x\Phi(x),$$

the GELU activation is represented by $Z_2 = \Phi(x_1) \otimes x_1$. Similarly, Shazeer's GEGLU layer corresponds to $Z_6 = \Phi(x_1) \otimes x_1 \otimes x_2$. Thus, extending the present study to $\phi = \Phi$ would recover both GELU and GEGLU within the same restricted enumeration.

GELU is also commonly approximated by a scaled Sigmoid gate,

$$\mathrm{GELU}(x) \approx x\sigma(1.702x).$$

This means that GELU-like behaviour is closely related to the Sigmoid-gated $Z_2$ form. When the affine projections inside and outside the gate are distinct, the learned projection weights may partially absorb this scalar factor. However, this approximation is not a substitute for explicitly evaluating $\phi = \Phi$, which we consider to be an interesting avenue for future work.

### 3.3 SinGLU

Of particular note is the function we call SinGLU, given by the equation:

$$\mathrm{SinGLU}(x, W_1, W_2, b_1, b_2) = \mathrm{Sin}(xW_1 + b_1) \otimes (xW_2 + b_2) \tag{8}$$

where $x$ is the input feature vector of length $[inputdim]$. $W_1$ and $W_2$ are learnable weight matrices of shape $[hiddendim \times inputdim]$, $b_1$ and $b_2$ are learnable bias vectors of length $[hiddendim]$.

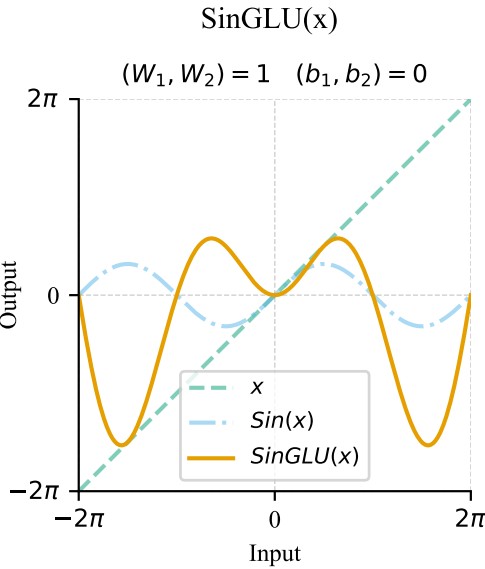

Figure 2: Input-output plot for our SinGLU layer function when all weights are 1 and all biases are 0, i.e. $\sin(x) \otimes x$. Dashed green lines shows $output = input$. Dash-dotted blue shows $output = \sin(input)$. SinGLU is effectively the product of these two.

The first weight and bias effectively control the frequency modulation and phase offset of the function's output. The second controls the amplitude. The input-output plot for a single neuron with weights equal to 1 and biases equal to 0 is shown in Figure 2. We show the effect of varying the parameters $W_1$, $W_2$, $b_1$, and $b_2$ in Figure 3.

Although Sin is a periodic activation, the resulting SinGLU layer is not generally periodic as a function of its input, because the sinusoidal gate is multiplied by the separate affine value pathway. It is periodic only in special cases, such as along input directions for which the value projection remains constant.

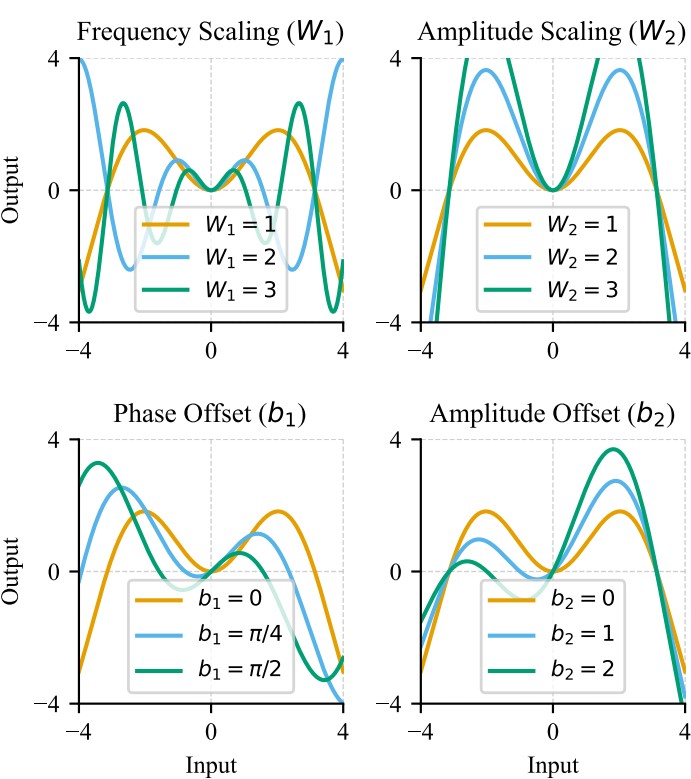

Figure 3: Scalar input-output plots for the SinGLU layer function $\text{SinGLU}(x) = \sin(xW_1 + b_1) \otimes (xW_2 + b_2)$ while varying $W_1$, $W_2$, $b_1$, and $b_2$ individually. The baseline setting uses $W_1 = W_2 = 1$ and $b_1 = b_2 = 0$; in each panel, one parameter is increased while the others are held fixed. Panels (a) and (b) show the effect of increasing $W_1$ and $W_2$, respectively, while panels (c) and (d) show the effect of increasing $b_1$ and $b_2$, respectively. In this scalar visualization, $W_1$ and $b_1$ change the frequency and phase of the sinusoidal gate, while $W_2$ and $b_2$ scale and shift the affine value pathway. In the full vector-valued layer, these effects occur component-wise through learned affine projections.

The role of the second bias term can be seen by distributing the Hadamard product over the value pathway:

$$\sin(xW_1 + b_1) \otimes (xW_2 + b_2) = \sin(xW_1 + b_1) \otimes xW_2 + b_2 \otimes \sin(xW_1 + b_1).$$

The second term shows that $b_2$ introduces an additional sinusoidal contribution that does not depend on the magnitude of the value projection $xW_2$. Thus, even when the value projection is close to zero, the output can contain a gated sinusoidal component scaled by $b_2$. The bias $b_1$ shifts the pre-activation of the sine gate, changing which input regions are amplified, suppressed, or sign-flipped by the gate.

### 3.4 Importance of Function Order in GLU-type Layers

We suspect that the relative differences in performance of different layer functions are likely to be primarily driven by the shapes of their input-output responses. We illustrate these differences in Figure 4. First order GLU-type functions with Sigmoid activations (SiLU, GLU) have outputs which tend towards linearity as inputs increase in magnitude. Second order functions with Sigmoid (SwiGLU, etc.) are approximately parabolic for large positive inputs. This is also true of GLU-type functions with Tanh activations. However, unlike Sigmoid, Tanh activated functions do not tend towards zero as inputs tend towards negative infinity. Amplitudes for Sin activated functions also increase linearly for first-order functions and parabolically for second-order functions.

Input-Output Plots of Feed-Forward Layer Functions

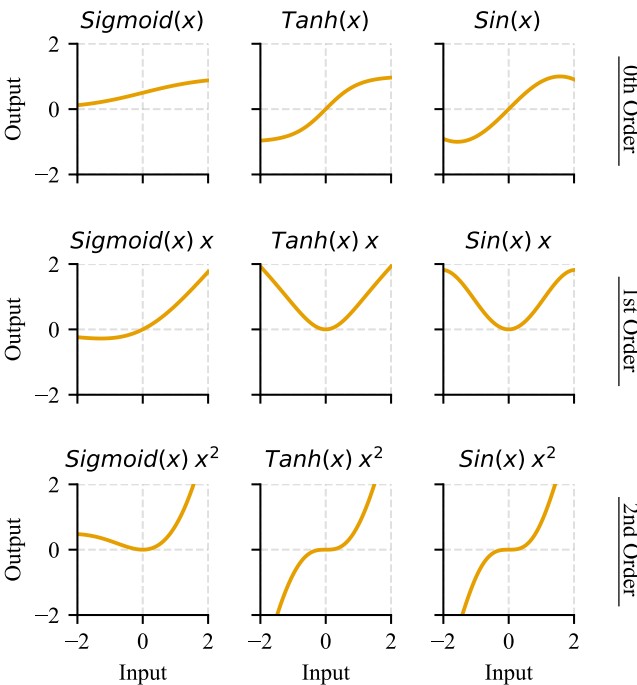

Figure 4: Input-output plots of $0^{th}$, $1^{st}$, and $2^{nd}$ order GLU-type layers with Sigmoid, Tanh, and Sin activation functions with all weights equal to 1 and no biases. Plots are arranged by function order row-wise and by activation function column-wise. It can be seen that for all function orders Tanh and Sin activated layers have very similar input output plots near zero.

### 3.5 Architecture

We implement 21 ViT-Tiny (Wu et al., 2022) models with the layer functions in the MLP blocks. Vision Transformers are typically thought to require vast amounts of training data to suppress large negative eigenvalues in the Hessian and stabilize training (Park & Kim, 2022). Despite this, ViT-Tiny (a ViT model with substantially reduced parameters) has been shown to work well on small image datasets (Gani et al., 2022). Further, pairing a small model with small datasets allows us to efficiently evaluate a wide range of models under limited compute.

Table 1: ViT-Tiny model architecture

| Parameter | Value |
|---|---|
| Width | 192 |
| Depth | 12 |
| MLP ratio (base) | 4 |
| MHSA heads | 3 |
| Parameters | 5.5 million |

In order to ensure a fair comparison between layer functions we reduce the size of the hidden dimension of the MLP block depending on the number of unique weight matrices in the layer function. As in Shazeer's implementation we reduce the hidden dimension to:

$$HiddenDim_{new} = \frac{2 \times HiddenDim}{n_{max} + 1} \tag{9}$$

Where $n_{max}$ is the number of unique weight matrices, i.e. 1 for $Z_2$, or 3 for $Z_7$ in Equation 7. This ensures the parameter count is constant between functions. To further reduce runtime and memory usage we combine the activation and Hadamard product operations via a fused kernel generated at runtime using PyTorch's JIT compiler.

## 3.6 Datasets

The datasets used in this study are CIFAR-10 (Krizhevsky et al., 2009), CIFAR-100 (Krizhevsky et al., 2009), SVHN (Netzer et al., 2011), and ImageNet64 (Chrabaszcz et al., 2017). These datasets have been widely used as benchmarks for small vision transformers (Park & Kim, 2022; Wu et al., 2022; Gani et al., 2022).

**CIFAR-10** consists of 60,000 colour images, each of size 32×32 pixels. These images span 10 object classes such as airplanes, cars, and birds. The dataset is divided into 50,000 training images and 10,000 test images. This gives 5,000 training images and 1,000 test images per class. CIFAR-10 serves as a widely recognized benchmark for machine learning and image classification models.

**CIFAR-100** also contains 60,000 colour images of size 32×32 pixels. However, these images are distributed across 100 classes. As such, there are only 500 training images and 100 test images per class. This makes CIFAR-100 useful in testing a model's capacity to learn generalizable functions when the problem space is sparsely sampled.

**SVHN** is a dataset of over 600,000 real-world images of house numbers taken from Google Street View. The images are 32×32 pixels and have one number at the centre of each image. However, almost all images show other numbers to the sides of the centred number. This adds an interesting extra challenge as the model must learn to ignore the numbers to the sides.

**ImageNet64** is a variant of the ImageNet1k dataset that has been resized to 64×64 pixels. ImageNet1k has over a million labelled images covering 1,000 distinct classes of natural objects such as cats, dogs, cars etc. ImageNet1k is considered a standard benchmark for machine learning image classification.

## 3.7 Training Regimes

We evaluate each of the 21 layer-function variants on CIFAR-10, CIFAR-100, and SVHN. Based on these controlled experiments, we identify SinGLU as a strong first-order alternative to SwiGLU and therefore perform an additional comparison between SinGLU and SwiGLU on ImageNet64.

The purpose of these experiments is not to establish state-of-the-art accuracy on these benchmarks. Instead, we use a fixed ViT-Tiny architecture and identical training recipes to isolate the relative effect of the MLP

layer function. Consequently, the relevant comparison is between layer functions under matched conditions, rather than against heavily optimized dataset-specific models.

As we are concerned only with relative performance between layer functions, we use the same hyperparameters for CIFAR-10, CIFAR-100, and SVHN, since these datasets are of comparable size and all contain 32×32 images. The hyperparameters are taken from Park & Kim (2022) and are reported in Table 2. The patch size used for CIFAR-10, CIFAR-100, and SVHN is 2×2. For ImageNet64, we double the patch size to 4×4 to account for the larger 64×64 image resolution.

Table 2: Training hyperparameters

| Parameter | Value |
|---|---|
| Patch size | 2×2 or 4×4 |
| Image padding | 4 pixels |
| Colour jitter | 0 |
| Batch size | 96 |
| Epochs | 300 |
| Warm-up epochs | 5 |
| Weight decay | 0.05 |
| Optimizer | AdamW |
| Learning rate | $1.25 \times 10^{-4}$ |
| LR scheduler | cosine annealing |
| LR $T_{\max}$ | 300 |
| Auto-augment policy | `rand-m9-n2-mstd1.0` |
| Label smoothing | 0.1 |
| Mixup $\alpha$ | 0.8 |
| CutMix $\alpha$ | 1.0 |
| Mixup probability | 1.0 |

## 4 Results and Discussion

In this section we present the Top-1 accuracy of all 21 models for the four datasets. Top-1 accuracy is the proportion of times the model's most confident prediction (the class with the highest predicted probability) matches the true class label. Formally, we can say that given a set of data instances N, with each instance $i$ having a true class label $y_i$, and a model prediction $\hat{y}_i$ representing the class with the highest confidence:

$$\text{Top-1 accuracy} = \frac{1}{N} \sum_{i=1}^{N} \left\{ \begin{array}{ll} 1, & \hat{y}_i = y_i \\ 0, & \text{otherwise} \end{array} \right. \tag{10}$$

Due to the scaling by $\frac{1}{N}$ the Top-1 accuracy function necessarily returns a value between 0 and 1. In the tables following we scale the values by a factor of 100 to give the accuracy as a percentage.

Besides the per-dataset results presented in the following tables, Figure 5 shows the mean test-set Top-1 accuracy over training for each evaluated layer function on CIFAR-10, CIFAR-100, and SVHN. The curves suggest that training behaviour often clusters by GLU function order, although the best-performing order depends on both the activation function and the dataset. This supports the view that GLU order and activation choice should be considered jointly, rather than treating the activation function alone as the determining factor.

The Tanh and Sin variants also show qualitatively similar training dynamics across datasets, both in the general shape of the curves and in the relative ordering of the different GLU orders. This observation is consistent with our motivation for testing Sin: early in training, Layer Normalization and near-zero initialization are likely to place many pre-activations in a regime where $\sin(x)$ and $\tanh(x)$ behave similarly. The

later divergence in performance between Tanh and Sin variants suggests that the non-monotonic behaviour of Sin away from zero may also affect the learned representations.

## Test-set Top-1 Accuracy (%) vs Training Epoch

Figure 5: Mean training curves showing test-set Top-1 accuracy versus training epoch for each evaluated GLU-type layer function on CIFAR-10, CIFAR-100, and SVHN. Curves are averaged over three random seeds and grouped by activation function and GLU order.

### 4.1 CIFAR-10

In Table 3 we report the final Top-1 accuracy for all models on the CIFAR-10 dataset. On CIFAR-10, SinGLU achieves the highest mean accuracy among the evaluated functions. From Figure 5 it may be seen that model training curves group by GLU-type function order. However, it is not the case that second-order functions outperform first- or zeroth-order functions in all cases. Instead, the best performing function order is activation dependent. For Sigmoid activations, second-order functions consistently outperform first order, which outperform zeroth. However, for Tanh and Sin activations first-order functions outperform second, which outperform zeroth.

Table 3: Final top-1 accuracy (%) on CIFAR-10 ($\uparrow$).

| GLU variant | Activation $\phi$ | | |
|---|---|---|---|
| | Sigmoid | Tanh | Sin |
| $\phi(x_1)$ | 85.24 $_{(0.34)}$ | 84.92 $_{(0.38)}$ | 86.46 $_{(0.17)}$ |
| $\phi(x_1) \otimes x_1$ | 88.78 $_{(0.37)}$ | 90.08 $_{(0.01)}$ | 89.90 $_{(0.05)}$ |
| $\phi(x_1) \otimes x_2$ | 89.35 $_{(0.13)}$ | **90.37** $_{(0.02)}$ | † **91.07** $_{(0.06)}$ † |
| $\phi(x_1) \otimes x_1 \otimes x_1$ | **90.62** $_{(0.21)}$ | 88.12 $_{(0.42)}$ | 88.68 $_{(0.61)}$ |
| $\phi(x_1) \otimes x_2 \otimes x_2$ | **90.67** $_{(0.22)}$ | 89.43 $_{(0.27)}$ | 88.95 $_{(0.36)}$ |
| $\phi(x_1) \otimes x_1 \otimes x_2$ | ⋆ 90.19 $_{(0.33)}$ ⋆ | 89.24 $_{(0.25)}$ | 89.07 $_{(0.12)}$ |
| $\phi(x_1) \otimes x_2 \otimes x_3$ | **90.49** $_{(0.20)}$ | 88.97 $_{(0.02)}$ | 88.91 $_{(0.27)}$ |

*Notes:* Shown are mean $_{(std)}$ of 3 training runs with random seeds.
†: Results for SinGLU. ⋆: Results for SwiGLU.
**Bold**: all means higher than SwiGLU. Underlined: highest mean overall.

## 4.2 CIFAR-100

In Table 4 we report the final Top-1 accuracy for all 21 models on the CIFAR-100 dataset. Although SinGLU achieves a higher mean accuracy than SwiGLU on CIFAR-100, it is not the best-performing function on this dataset. Instead the original GLU achieves the highest mean accuracy, suggesting that the preferred GLU-type function may depend on dataset density and overfitting behaviour.

Table 4: Final top-1 accuracy (%) on CIFAR-100 ($\uparrow$).

| GLU variant | Activation $\phi$ | | |
|---|---|---|---|
| | Sigmoid | Tanh | Sin |
| $\phi(x_1)$ | 65.00 $_{(0.32)}$ | 63.75 $_{(0.42)}$ | **65.99** $_{(0.03)}$ |
| $\phi(x_1) \otimes x_1$ | **67.28** $_{(0.80)}$ | **65.74** $_{(0.03)}$ | 65.19 $_{(0.02)}$ |
| $\phi(x_1) \otimes x_2$ | **67.63** $_{(0.44)}$ | 65.07 $_{(0.48)}$ | † **66.06** $_{(0.65)}$ † |
| $\phi(x_1) \otimes x_1 \otimes x_1$ | 65.21 $_{(0.53)}$ | **66.10** $_{(0.49)}$ | **66.01** $_{(0.41)}$ |
| $\phi(x_1) \otimes x_2 \otimes x_2$ | 64.87 $_{(0.32)}$ | 64.10 $_{(0.39)}$ | 64.32 $_{(0.45)}$ |
| $\phi(x_1) \otimes x_1 \otimes x_2$ | ⋆ 65.31 $_{(0.29)}$ ⋆ | 64.37 $_{(0.35)}$ | 64.71 $_{(0.37)}$ |
| $\phi(x_1) \otimes x_2 \otimes x_3$ | 65.07 $_{(0.57)}$ | 63.42 $_{(0.47)}$ | 63.32 $_{(0.43)}$ |

*Notes:* Shown are mean $_{(std)}$ of 3 training runs with random seeds.
†: Results for SinGLU. ⋆: Results for SwiGLU.
**Bold**: all means higher than SwiGLU. Underlined: highest mean overall.

In Figure 5 we present the training curves for CIFAR-100. These results are inconsistent with CIFAR-10, where the top-performing function order was activation dependent. Instead $1^{st}$ and $2^{nd}$ order functions perform poorly for all activations. We suspect that these results indicate that more complex functions generalize poorly when the problem space is sparsely sampled. We explore this further in Section 4.6. If we are correct and the poor results for higher order functions are due to increased propensity for overfitting, this could likely be overcome by stronger regularization, for example by increasing the weight decay or using dropout.

## 4.3 Street View House Numbers

In Table 5 we report the final Top-1 accuracy for all 21 models on the SVHN dataset. As with CIFAR-10 the model using the SinGLU function had the highest mean accuracy. It should be noted that SVHN has the same number of classes as CIFAR-10, and slightly more images per class. It is also an arguably easier task, implying that the problem space is more densely sampled than CIFAR-10.

Figure 5 shows the training curves for SVHN. These curves can be seen to be similar to those of CIFAR-10. As before, the curves generally couple based on GLU-type function order. Similarly we can see that, for Sigmoid activations, second-order functions consistently outperform first order, which outperform zeroth.

Again, as with CIFAR-10, for Tanh and Sin activations first-order functions are consistently best, followed by second order, then zeroth.

Table 5: Final top-1 accuracy (%) on SVHN ($\uparrow$).

| GLU variant | Activation $\phi$ | | |
|---|---|---|---|
| | Sigmoid | Tanh | Sin |
| $\phi(x_1)$ | 94.97 $_{(0.27)}$ | 95.28 $_{(0.30)}$ | 96.07 $_{(0.43)}$ |
| $\phi(x_1) \otimes x_1$ | 96.52 $_{(0.10)}$ | 97.18 $_{(0.12)}$ | 97.18 $_{(0.19)}$ |
| $\phi(x_1) \otimes x_2$ | 96.82 $_{(0.04)}$ | **97.36** $_{(0.14)}$ | † **97.36** $_{(0.09)}$ † |
| $\phi(x_1) \otimes x_1 \otimes x_1$ | **97.33** $_{(0.10)}$ | 97.07 $_{(0.18)}$ | 97.00 $_{(0.12)}$ |
| $\phi(x_1) \otimes x_2 \otimes x_2$ | **97.29** $_{(0.05)}$ | 97.25 $_{(0.11)}$ | 96.96 $_{(0.12)}$ |
| $\phi(x_1) \otimes x_1 \otimes x_2$ | ⋆ 97.27 $_{(0.15)}$ ⋆ | 97.13 $_{(0.17)}$ | 97.11 $_{(0.17)}$ |
| $\phi(x_1) \otimes x_2 \otimes x_3$ | 97.09 $_{(0.16)}$ | 97.17 $_{(0.21)}$ | 97.08 $_{(0.05)}$ |

*Notes:* Shown are mean $_{(std)}$ of 3 training runs with random seeds.
†: Results for SinGLU. ⋆: Results for SwiGLU.
**Bold**: all means higher than SwiGLU. Underlined: highest mean overall.

## 4.4 ImageNet64

Shown in Table 6 are the Top-1 accuracies on ImageNet64 for SwiGLU and SinGLU. ImageNet64 was run once due to computational constraints and is therefore reported without a standard deviation. This result should be treated as supporting evidence rather than a statistically validated comparison. Consistent with the results on previous datasets, SinGLU again has a higher top-1 accuracy.

Table 6: Final top-1 accuracy ($\uparrow$) on ImageNet64. The best result is shown in bold and underlined.

| Layer function | Results |
|---|---|
| $SwiGLU$: $\sigma(x_1) \otimes x_1 \otimes x_2$ | 66.47 |
| $SinGLU$: $\sin(x_1) \otimes x_2$ | **67.73** |

## 4.5 Computation Time Comparisons

As previously discussed, we maintain the parameter count between models by resizing the MLP hidden layers based on the number of weight matrices. However, this does not in itself guarantee equal computation time between models. We therefore benchmark the time taken to compute a forward pass.

We create models with patch size 2 and 10 output logits, in line with our training regime for all datasets except ImageNet64. We then generate 10 tensors of Gaussian noise of size $[128 \times 3 \times 32 \times 32]$. For each tensor we randomly initialize the models and perform 20 forward passes on an NVIDIA A100 GPU, recording the time taken for the final 10. This allows a warm-up period for the timings to stabilize. For each model we take the mean across all forward passes and present these results in Table 7. We highlight the time taken for the SinGLU model with daggers and the SwiGLU model with stars.

It may be seen that there is less than 1% difference between activations (Sigmoid, Tanh, Sin) for any given layer function. However, there is a significant difference between $0^{th}$ order and $1^{st}$ order functions (maximum difference of 6.42% for Sigmoid, 6.42% for Tanh, and 6.24% for Sin). There is a more negligible difference between $1^{st}$ and $2^{nd}$ order functions (maximum difference of 1.60% for Sigmoid, 1.48% for Tanh, and 1.45% for Sin). It is likely that the difference in computation time between functions could be minimized by writing custom CUDA kernels which fuse operations. We leave this as an avenue for future work. It should be noted that there is less than 0.1% difference in computation time for SwiGLU and SinGLU models.

Table 7: Computation time per forward pass (milliseconds) for ViT-Tiny layers. Daggers (†) highlight the results for SinGLU; stars (⋆) highlight SwiGLU.

| Layer function | Activation $\phi$ | | |
|---|---|---|---|
| | Sigmoid | Tanh | Sin |
| $\phi(x_1)$ | 30.31 | 30.30 | 30.37 |
| $\phi(x_1) \otimes x_1$ | 32.39 | 32.38 | 32.39 |
| $\phi(x_1) \otimes x_2$ | 31.98 | 32.01 | $^{\dagger}32.03^{\dagger}$ |
| $\phi(x_1) \otimes x_1 \otimes x_1$ | 32.50 | 32.49 | 32.51 |
| $\phi(x_1) \otimes x_2 \otimes x_2$ | 32.06 | 32.09 | 32.11 |
| $\phi(x_1) \otimes x_1 \otimes x_2$ | $^{\star}32.06^{\star}$ | 32.09 | 32.10 |
| $\phi(x_1) \otimes x_2 \otimes x_3$ | 32.30 | 32.34 | 32.36 |

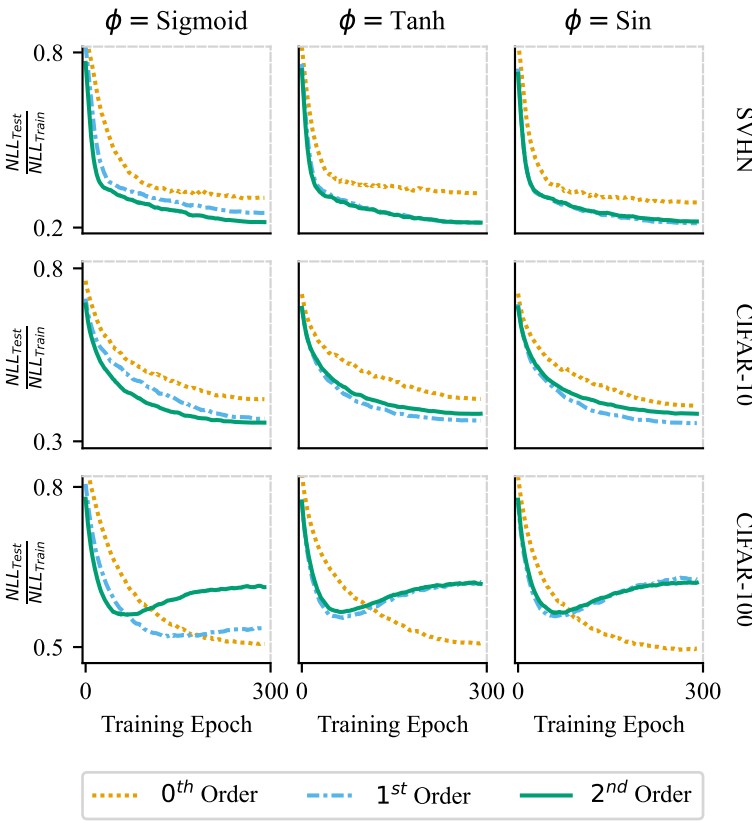

Figure 6: Plots of $NLL_{Test}/NLL_{Train}$ vs Training Epoch for the datasets SVHN, CIFAR-10, and CIFAR-100. Presented are the means across all functions of a given order and activation. The curves have been smoothed using a rolling mean with a window of ten epochs. Downward trends indicate improving generalization, upward trends indicate increasing overfitting. Downward trends are seen for all models except $1^{st}$ and $2^{nd}$ order functions on CIFAR-100.

### 4.6 Overfitting

In Section 4.2 we suggested that the poor performance of first- and second-order GLU-type functions on the CIFAR-100 dataset could be due to the sparsely sampled problem space. In other words, we suggest that more expressive, complex functions could be more prone to overfitting on datasets with few examples per class. To explore this hypothesis we compared the Negative Log-Likelihood (NLL) of the models on the training and test sets at every epoch for the SVHN, CIFAR-10, and CIFAR-100 datasets. We visualize these results in Figure 6.

Dividing the $NLL_{Test}$ by $NLL_{Train}$ gives a measure of overfitting. Values of $NLL_{Test}/NLL_{Train}$ greater than 1 indicate overfitting. While none of the models had values greater than 1 at any point it is important to remember that we perform heavy augmentation of the training set making the training set significantly harder than the test set. More important is the slope of the curve of $NLL_{Test}/NLL_{Train}$ over time. A downward trend indicates the model is generalizing better over time, while an upward trend indicates the model is becoming increasingly overfit.

From Figure 6 it may be seen that all functions experience downward trends in $NLL_{Test}/NLL_{Train}$ over time except for $1^{st}$ and $2^{nd}$ order functions on the CIFAR-100 dataset. This suggests that the poor performance of $1^{st}$ and $2^{nd}$ order functions on CIFAR-100 is due to increased overfitting relative to the other functions. This is consistent with our hypothesis that more expressive, complex functions are more prone to overfitting on sparsely sampled problem spaces.

## 5 Conclusion

In this study we systematically investigated the impact of GLU-type nonlinear layer functions in the MLP portions of Vision Transformers. We tested these layer functions with three different activation functions: Sigmoid, Tanh, and Sin. From these we identified SinGLU, a $1^{st}$ order GLU-type layer with a Sin activation as the most performant on the CIFAR-10 and SVHN datasets. A single-run ImageNet64 comparison provides additional supporting evidence that SinGLU can outperform SwiGLU beyond the smaller $32{\times}32$ datasets, although further multi-seed experiments are required before drawing strong statistical conclusions. However, our testing on CIFAR-100 suggests that more expressive functions such as SinGLU and SwiGLU may be more prone to overfitting on sparsely sampled problem spaces. We suggest that this could be mitigated by stronger regularization such as increased weight decay or dropout, or by heavier data augmentation.

Our activation comparison was restricted to Sigmoid, Tanh, and Sin. Although this includes the original GLU and SwiGLU within our notation, it does not exhaustively reproduce all GLU variants considered by Shazeer, such as ReGLU and GEGLU. Extending the present enumeration to additional activations is an important direction for future work.

We showed that different order GLU-type layers have fundamentally different dynamics, with $1^{st}$ order functions having a pronounced linear quality and $2^{nd}$ order functions having a pronounced parabolic quality. We further showed that this causes different layer functions of the same order to have similar training curves. However, the most performant order depends on the activation being used, with Sin and Tanh showing best results with $1^{st}$ order GLU functions, and Sigmoid showing best results with $2^{nd}$ order functions.

We believe we have demonstrated that periodic activation functions such as Sin may be used as gating functions in Vision Transformers without explicit changes to the architecture or training methods. Our working hypothesis for SinGLU's performance is twofold. First, the combination of Pre-Layer-Normalization and near-zero weight initialization confines the sinusoidal gate to its near-monotonic regime at initialization, avoiding the optimization pathologies historically associated with periodic activations. Second, as training progresses the network can access the non-monotonic structure of sin where useful, granting SinGLU a larger functional repertoire than monotonic gates of comparable smoothness. Whether this fully accounts for the observed gains, or whether other mechanisms such as strengthening of a soft inductive bias towards low-frequency solutions under weight decay also contribute, remains an open question for future work.

While our results suggest an advantage for SinGLU over SwiGLU under controlled ViT-Tiny baseline conditions, we acknowledge several limitations. First, due to computational constraints, our experiments were

restricted to the ViT-Tiny architecture (approximately 5.5 million parameters) with proportionally small datasets such as CIFAR-10 and ImageNet64. It remains an open question whether the performance benefits of periodic gating mechanisms generalize to massively scaled models such as ViT-Huge or Large Language Models. Second, to ensure a strictly controlled comparison, we applied an identical training recipe and hyperparameter set across all 21 models. It is entirely possible that SwiGLU and other higher-order functions might achieve better relative performance if hyperparameters such as learning rate, weight decay, and initialization schemes were independently optimized for each specific layer function. Future work should explore the scalability of SinGLU on larger architectures and datasets, alongside function-specific hyperparameter tuning.

### Acknowledgments

This work was supported by EPSRC DTP grant number: EP/W524670/1. All results were obtained using the ARCHIE-WeSt High Performance Computer (www.archie-west.ac.uk) based at the University of Strathclyde.

### Reproducibility Statement

All code and configuration files required to reproduce our results are available at `https://github.com/Luke-Byrne-Eng/SinGLU-Experimental-Code`.

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
