# OpenReview forum: "SinGLU: Sinusoidal Gated Linear Units Improve Classification Accuracy of Small Vision Transformers"
_TMLR — Accepted by TMLR_

### Review · Reviewer_8KLG · 2026-04-07

**Summary Of Contributions:**

This paper investigates the impact of various GLU variants on small ViT architectures for image classification. The authors evaluate three activation functions, namely Sigmoid, Tanh, and Sin, across 0th to 2nd-order GLUs. Their findings on the CIFAR-10 and SVHN datasets indicate that the first-order GLU combined with the Sin activation function achieves the highest performance.

**Additional Comments:**

In its current state, the paper requires a more coherent narrative and a clearer line of argumentation. Additionally, the presentation of results in the tables is suboptimal, making them difficult to interpret. I strongly encourage the authors to address these points and incorporate all the feedback provided in the 'Requested Changes' section.

**Audience:**

Yes

**Audience Explanation:**

The authors provide a comprehensive study on the influence of different activation functions that follow the general form of a GLU for small classification ViTs. While these results are not particularly groundbreaking, they are presented in a solid manner and under a fair comparison.

**Broader Impact Concerns:**

N/A.

**Claims And Evidence:**

No

**Claims Explanation:**

The paper contains several overgeneralized or incorrect claims that lack clear and convincing evidence. To ensure technical accuracy, the authors must rewrite these parts of the manuscript to more accurately as described in the Requested Changes.

**Requested Changes:**

These are the requested changes, the ordering does not reflect the importance of each point:

1.	The manuscript oversimplifies Transformer architectures and fails to include necessary details e.g. regarding positional embeddings.
2.	The authors overstate the role of the activation function by labeling it the 'only' source of non-linearity. In a standard Transformer, non-linearity is introduced both through the feed-forward activation layers and the Softmax normalization applied during the attention calculation.
3.	The presence and role of the bias vector within the GLU formulation require clarification.
4.	“In both GLU and Shazeer’s GLU variants the weight matrices and bias vectors for the activated and nonactivated neurons are distinct and different.“ True?!
5.	The "zeroth-order" variant functions as a standard activation function rather than a gating mechanism; this distinction should be made more clear.
6.	The figure sequencing is inconsistent, as Figure 1 is shown before Figure 2 but Figure 2 is referenced first in the text.
7.	The motivation for selecting the Sin function as an additional activation is not sufficiently explained or justified.
8.	The conclusion that SinGLU performs best is stated before the supporting results are presented, disrupting the logical flow of the paper.
9.	In Figure 4, the explanation of how the bias term $b_2$ results in an amplitude offset remains unclear.
10.	The claim of a "systematic enumeration" of all zeroth to second-order GLU-type functions is an overstatement, as the study only evaluates a subset of three activations.
11.	The discussion regarding "periodic activation functions" is potentially inaccurate given that GLUs are not periodic by design.
12.	“Park has also shown that the MHSA block“ here the citation missing.
13.	“Online normalization standardizes the data distribution before applying a learned linear affine.“ here a word is missing and the spacing is broken.
14.	The section on Online Normalization does not seem relevant to the paper’s primary focus.
15.	Figure 5 is poorly designed; it is not completely clear what is compared, blue and green seem to be the sin activation function, but orange is what? Than for a normal distribution? Is it the same as for blue?
16.	The authors report only mean values without standard deviations, making it impossible to determine the statistical significance of the results.
17.	The reported performance metrics are notably low when compared to current state-of-the-art results on the utilized datasets.
18.	On the CIFAR-100 dataset, the claim that SinGLU is superior does not hold, as it performs significantly worse than the alternatives.
19.	The claim that SinGLU outperforms SwiGLU by a "statistically significant margin" on ImageNet64 is incorrect and contradicted by the results provided.
20.	The study would be significantly more comprehensive and complete if it included a full comparison with the various GLU variants proposed by Shazeer (2020).

---

> ### Author Response · Authors · 2026-05-15
> **Initial response to reviewer 8KLG**
>
> We are grateful to Reviewer 8KLG for their thorough review, which has substantially improved the manuscript's clarity, precision, and flow. We agree on all points, and have addressed them below.
>
> **Points 1 & 2: Transformer architecture details and sources of non-linearity.**
>
> The Introduction and background now describe patch embeddings, positional embeddings, residual connections, and Layer Normalization. We have also corrected the text to identify the softmax in attention and Layer Normalization as additional sources of non-linearity alongside the MLP activation.
>
> **Points 3 & 4: Bias vectors and the "distinct and different" claim.**
>
> We have corrected this and rewritten Section 3.1 to explain that while Shazeer's variants omit biases following the T5 codebase, we adopt a generalized notation including biases, since the original GLU used them and they are relevant for phase and amplitude modulation in sinusoidal gating.
>
> **Point 5: Zeroth-order variant clarification.**
>
> Section 3.2 now explicitly states that $Z_1 = \phi(x_1)$ is not a gating mechanism but is included as a baseline, allowing us to separate the effect of the activation from the effect of gating.
>
> **Point 6: Figure sequencing.**
>
> Corrected, thank you.
>
> **Point 7: Motivation for Sin.**
>
> Section 2.4 has been substantially expanded. Layer Normalization and standard initialization places pre-activations near zero where $\sin(x) \approx \tanh(x)$. As training progresses, the network can optionally exploit the non-monotonic structure of $\sin$.
>
> **Point 8: Logical flow.**
>
> All claims based on results now follow the supporting evidence. The abstract, introduction, results, and conclusion now use measured language.
>
> **Point 9: Bias term and amplitude offset in Figure 4.**
>
> We have added an explicit algebraic decomposition:
> $$\sin(xW_1+b_1)\otimes(xW_2+b_2) = \sin(xW_1+b_1)\otimes xW_2 + b_2\otimes \sin(xW_1+b_1),$$
> which makes clear that $b_2$ contributes an additional sinusoidal term, while $b_1$ shifts the pre-activation of the gate.
>
> **Point 10: "Systematic enumeration" overstatement.**
>
> We have updated the contributions list and abstract with softened language, clarifying that we are performing a restricted enumeration of affine-projection sharing patterns, while Sigmoid, Tanh, and Sin represent a targeted range of activations.
>
> **Point 11: "Periodic activation functions" terminology.**
>
> Corrected throughout. Section 2.4 and the SinGLU definition now explicitly state that SinGLU is not generally periodic in its input.
>
> **Points 12 & 13: Missing Park citation and Online Normalization typos.**
>
> We have folded the relevant claim into the preceding reference and corrected the missing word and spacing.
>
> **Point 14: Relevance of the Online Normalization section.**
>
> This section has been substantially shortened. We retained only the material that supports our argument in Section 2.4 that pre-Layer-Normalization constrains the input distribution to the sinusoidal gate.
>
> **Point 15: Figure 5 design.**
>
> On reflection, Figure 5 was not substantially helping the argument it was intended to support. We have removed it and developed the underlying argument as prose in Section 2.4.
>
> **Point 16: Standard deviations.**
>
> We now report mean and standard deviations for all experiments except ImageNet64.
>
> **Point 17: Performance metrics vs. SOTA.**
>
> Our hyper-parameters were taken from the literature, and our SwiGLU baseline achieves 90.19% / 65.31% / 97.27% on CIFAR-10 / CIFAR-100 / SVHN, within roughly one point of a widely-used reference implementation (omihub777/ViT-CIFAR on GitHub: 90.92 / 66.54 / 97.31). Section 3.6 has been updated to clarify that the study focuses on the relative effect of the MLP layer function under matched conditions, rather than targeting SOTA.
>
> **Point 18: CIFAR-100 claim.**
>
> The CIFAR-100 section now more explicitly acknowledges that the original GLU is the best-performing function on this dataset. However, SinGLU still exceeds SwiGLU and we believe the poor performance of the higher order functions is largely explained by our overfitting analysis.
>
> **Point 19: ImageNet64 "statistical significance" claim.**
>
> We have removed this language, thank you.
>
> **Point 20: Full comparison with Shazeer (2020) variants.**
>
> We agree that a full comparison would strengthen the paper. However, we were unable to include these comparisons within the available compute budget and timeframe. We have updated Section 3.2 to note that within our notation, GELU is recovered as $Z_2 = \Phi(x_1)\otimes x_1$ with $\Phi$ the standard normal CDF, and GEGLU as $Z_6 = \Phi(x_1)\otimes x_1\otimes x_2$, so extending our enumeration to $\phi = \Phi$ would naturally recover both. In the Conclusion we state that an exhaustive evaluation of Shazeer's full variant set is an important direction for future work.
>
> Again, we would like to express our thanks for reviewer 8KLG's detailed and constructive criticism, as we believe it has substantially improved the work.

---

### Review · Reviewer_WfMs · 2026-04-11

**Summary Of Contributions:**

This paper studies various gating mechanisms for small vision transformers. Given basic affine module $x_i=xW_i+b_i$, it considers gating up to 2nd orders, i.e., $\phi(x_1)$, $\phi(x_1)\otimes x_1, \phi(x_1)\otimes x_2$ and $\phi(x_a)\otimes x_b\otimes x_c$ for $a, b, c\in [3]$. The activation functions $\phi$ tried are sigmoid, tanh and sin. Experiments are performed with a ViT of size 5.5M parameters and on datasets CIFAR-10, CIFAR-100, SVHN and Imagenet64. Results are mixed, but maybe the major message is that for sigmoid, 2nd order gating generally works well, while for tanh and sin, first order gating is enough. They also conducted experiments on overfitting by comparing the ratio between NLL of test loss and NLL of training loss, and CIFAR-100 has its first order and second order diverges.

**Additional Comments:**

Some typos:

* Section 1, third paragraph, 'GLU-type' and 'Gated Linear Units' should be written as \`GLU-type' and \`Gated Linear Units'. In general, ' ' in latex should be typed as \` '.

* Section 2.3, first paragraph, "applying a learned linear affine" -> "applying a learned affine transform".

* Page 6, last paragraph, "In Figure 5 A global minimum" -> "In Figure 5, a global minimum".

* Section 3.1, above Equation (7), "The output Z" -> "The output $Z$".

**Audience:**

Yes

**Audience Explanation:**

I'm not sure on this one, as I'm not an expert in vision in general, but I would imagine the experimental results are interesting to some ViT audience.

**Broader Impact Concerns:**

N/A.

**Claims And Evidence:**

Yes

**Claims Explanation:**

The paper makes very little claims, it mostly presents results, and discusses some possibilities why that's the case.

**Requested Changes:**

I'm not an expert in ViT and vision, but I feel the results of this paper are quite thin. Essentially, the authors simply try different gating mechanisms on relatively small models. While the experiments are "extensive", I feel some of them are redundant, e.g., I believe $\phi(x_1)\otimes x_2\otimes x_3$ has the same expressive power as $\phi(x_1)\otimes x_2$, and $\phi(x_1)\otimes x_2\otimes x_2$ seems strictly weaker than its first order counterpart. Also, I'm not sure one could conclude that first order sin activation is any useful in practice, due to the small size of the experiments. Overall, I feel I didn't learn too much from the paper.

---

> ### Author Response · Authors · 2026-05-15
> **Initial response to reviewer WfMs**
>
> We thank Reviewer WfMs for their careful reading and for raising the question of redundancy between layer variants, which has helped us improve Section 3.2. We address the reviewer's specific comments below.
>
> **Comment 1: Results feel thin; doubts about small model size.**
>
> We chose ViT-Tiny to isolate the effect of the MLP layer function under matched conditions: at this scale, 21 variants can be trained with multiple seeds per dataset under identical recipes. We have clarified this intentional scope in Sections 1 and 3.6, and explicitly listed extension to larger architectures as a limitation in the Conclusion. We agree that the contribution is empirical and intentionally scoped, and have softened the abstract, introduction, and conclusions to reflect this.
>
> **Comment 2: Redundancy of layer functions.**
>
> We thank the reviewer for raising this. The reviewer is correct that some variants share algebraic structure, but they are not generally redundant in their induced input interactions. In response, we have expanded Section 3.2 with the following points:
>
> - $Z_2 = \phi(x_1)\otimes x_1$ and $Z_4 = \phi(x_1)\otimes x_1\otimes x_1$ reuse the same affine projection, but the additional Hadamard factor in $Z_4$ introduces a quadratic dependence in $x_1$ that changes the resulting function and gradients.
> - Variants with more independent projections introduce bilinear interactions. Expanding the non-activated component of $Z_7 = \phi(x_1)\otimes x_2\otimes x_3$ gives $(xW_2+b_2)\otimes(xW_3+b_3)$, which contains pairwise multiplicative interactions between two independent affine projections of the input. These interactions are absent from $Z_3$.
>
> The effects of this quadratic behaviour are seen in the bottom row of Figure 4 in our revised manuscript, which illustrates the polynomial growth characteristic of the second-order GLU variants.
>
> Whether these structural differences translate into measurable accuracy differences is precisely the empirical question motivating the enumeration.
>
> **Comment 3: Typos and LaTeX formatting.**
>
> Thank you for highlighting this. We have corrected the noted typos, fixed the quotation mark formatting throughout, and corrected the mathematical notation.
>
>
> We thank the reviewer again for their careful reading and constructive criticism. Their comments helped us clarify the purpose of the enumeration, better distinguish structural from empirical redundancy, and more accurately present the scope of the study.

---

### Review · Reviewer_pcFB · 2026-05-04

**Summary Of Contributions:**

The paper proposes a variant of Gated Linear Unit in Vision Transformers. The effectiveness of SinGLU is validated across various datasets, e.g., CIFAR-10, CIFAR-100, and ImageNet-64. Some interesting curves can further inspire the research of the community, e.g., Figure 5.

**Audience:**

Yes

**Audience Explanation:**

The architecture of Transformers is an attractive topic in the comunity of machine learning. If researchers can gain some insights of the designing, the paper is valuable.

**Claims And Evidence:**

Yes

**Claims Explanation:**

The claims are suported by accurate evidences, i.e., empirical validation results.

**Requested Changes:**

Negatives:
1. What is the core insight behind the designing of SinGLU? Is the introduction of Sine function? The computation of SinGLU is simpler than SwiGLU. Which parts lead to the performance gain of SinGLU?
2. Tables in Section 4 are expected to display clearly. Especially the denoting of SinGLU and SwiGLU.
3. What is the motivation to design SinGLU? Why do you introduce Sine function? More detailed analysis is expected.

---

> ### Author Response · Authors · 2026-05-15
> **Initial response to reviewer pcFB**
>
> We thank Reviewer pcFB for their constructive feedback and for noting that the empirical validation is convincing. We address each of their concerns below:
>
> **Comment 1 & 3: Core insight and motivation for SinGLU / Sin**
>
> We agree that the original draft did not articulate this clearly enough. Our motivation for including Sin follows a logical progression: Sigmoid is the canonical gating activation of GLU; Tanh is a recentered and rescaled Sigmoid bounded to $[-1,1]$; and Sin matches Tanh near the origin ($\sin(x) \approx \tanh(x)$) but becomes non-monotonic outside this regime. Including Sin therefore tests whether the normalization, initialization, regularisation, etc. of modern Vision Transformers make a bounded, non-monotonic gate viable within GLU-type MLP layers.
>
> We have expanded Section 2.4 to develop this argument. We have also added a paragraph to the Results section discussing the similarity of training curves in Figure 5 between functions using Tanh and Sin gates. This qualitative result is consistent with our hypothesis that early stage training dynamics would be similar between Tanh and Sin.
>
> We have also added a concrete two-part hypothesis to the Conclusion regarding the viability of sinusoidal gates in ViT-Tiny: (i) the avoidance of optimization pathologies at initialization, and (ii) the larger functional repertoire available once weights grow. However, we acknowledge that a mechanistic ablation isolating the specific source of SinGLU's gain remains outside the scope of this study, and we flag this explicitly as future work.
>
> **Comment 2: Table clarity.**
>
> We have redesigned all results tables in Section 4 with explicit "Notes" footers defining the dagger, star, bold, and underline conventions used to identify SinGLU, SwiGLU, results exceeding SwiGLU, and the overall best result, respectively.
>
> We thank the reviewer again for their constructive feedback. Their comments helped us clarify the motivation for SinGLU, improve the presentation of the results, and more carefully distinguish empirical observations from mechanistic claims.

---

### Review · Reviewer_9arX · 2026-05-15

**Summary Of Contributions:**

The paper systematically studies a restricted family of GLU-type MLP layers in small Vision Transformers by varying the multiplicative order and the gate activation, and proposes SinGLU, which uses a sinusoidal gate. Under matched parameter counts and training settings, SinGLU outperforms SwiGLU on several ViT-Tiny image classification benchmarks with nearly identical inference latency. The main strength is the clean controlled comparison, but the novelty is limited, the evidence is mostly empirical, and the experiments are restricted to small models/datasets with missing key baselines such as GELU, GEGLU, and ReGLU.

**Audience:**

Yes

**Audience Explanation:**

Some TMLR readers interested in Transformer architecture design, activation functions, and GLU variants would likely find the paper’s controlled comparison useful, especially the observation that sinusoidal gating can work stably in small pre-norm ViTs. However, the audience may be somewhat narrow because the current findings are limited to small vision models and do not yet establish broad relevance to larger ViTs or language Transformers.

**Claims And Evidence:**

No

**Claims Explanation:**

The claims are supported by controlled experiments showing that SinGLU outperforms SwiGLU in the reported ViT-Tiny settings, with matched parameter counts and similar latency. However, the evidence is not fully convincing because the experiments are limited to small models and datasets, ImageNet64 is only run once, and important baselines such as GELU, GEGLU, and ReGLU are missing. The mechanistic explanation for why sinusoidal gating helps is also plausible but not directly validated.

**Requested Changes:**

1. Add key baselines such as GELU, GEGLU, ReGLU, and standard ViT MLP.
2. Run ImageNet64 with multiple seeds and report variance.
3. Test on at least one larger or different ViT architecture.

---

### Decision · Action_Editor_Tehw · 2026-06-19

**Recommendation:** Accept as is

**Audience:**

Yes

**Audience Explanation:**

Yes.

Three reviewers agreed the controlled comparison is of interest to readers working on Transformer architecture design, activation functions, and GLU variants.

The finding that a bounded, non-monotonic sinusoidal gate trains stably and competitively in pre-norm ViT-Tiny MLPs is a useful, reproducible data point, and figures such as the training-curve comparison were noted as potentially inspiring further study.

Relevance to larger ViTs and language models is not yet established, but TMLR's bar is of interest to some segment of its audience, which this clean, matched-condition study clearly meets.

**Claims And Evidence:**

Yes

**Claims Explanation:**

The submission's central claim is narrow and well-matched to its evidence: 1) under-matched parameter counts and identical ViT-Tiny training recipes, 2) SinGLU attains higher mean accuracy than SwiGLU across CIFAR-10/100, SVHN, and ImageNet-64, with negligible latency difference.

The revision reports standard deviations over three seeds (except ImageNet-64, now framed as single-run supporting evidence), corrects the earlier overstated significance and enumeration claims, and softens conclusions appropriately.

Remaining limitations: 1) missing GELU/GEGLU/ReGLU baselines, 2) small-scale models, are acknowledged as scoped future work rather than overclaimed.

The claims as stated are accurate, clearly presented, and supported.